# Dual Interactions of Amphiphilic Gelatin Copolymer and Nanocurcumin Improving the Delivery Efficiency of the Nanogels

**DOI:** 10.3390/polym11050814

**Published:** 2019-05-07

**Authors:** Dinh Trung Nguyen, Van Thoai Dinh, Le Hang Dang, Dang Nam Nguyen, Bach Long Giang, Cong Truc Nguyen, Thi Bich Tram Nguyen, Le Van Thu, Ngoc Quyen Tran

**Affiliations:** 1Institute of Research and Development, Duy Tan University, Da Nang City 550000, Vietnam; trungnd.iams@gmail.com (D.T.N.); nguyendangnam@dtu.edu.vn (D.N.N.); 2Institute of Applied Materials Science, VAST, TL29, ThanhLoc Ward, Dist. 12, Ho Chi Minh City 700000, Vietnam; thoaidinh106@yahoo.com (V.T.D.); lehadang0804@gmail.com (L.H.D.); congtruc2205@gmail.com (C.T.N.); 3Graduate University of Science and Technology, VAST, TL29, Thanh Loc Ward, Dist. 12, Ho Chi Minh City 700000, Vietnam; 4School of Biotechnology, International University, Vietnam National University, Ho Chi Minh City 700000, Vietnam; 5NTT Hi-Tech Institute, Nguyen Tat Thanh University, 300A Nguyen Tat Thanh, Ward 13, District 4, Ho Chi Minh City 700000, VietNam; 6Department of Natural Science, Thu Dau Mot University, Thu Dau Mot City 590000, Vietnam; tramntb@tdmu.edu.vn

**Keywords:** nanomaterials, pluronic conjugated gelatin (GP) nanogel, nanocurcumin, nanogels, delivery

## Abstract

Herein, a new process to manufacture multicore micelles nanoparticles reinforced with co-assembly via hydrophobic interaction and electrostatic interaction under the help of ultrasonication was developed. The precise co-assembly between negative/hydrophobic drug and positive charged amphiphilic copolymer based pluronic platform allows the formation of complex micelles structures as the multicore motif with predefined functions. In this study, curcumin was selected as a drug model while positively charged copolymer was based on a pluronic-conjugated gelatin with different hydrophobicity length of Pluronic F87 and Pluronic F127. Under impact of dual hydrophobic and electrostatic interactions, the nCur-encapsulated core–shell micelles formed ranging from 40 nm to 70 nm and 40–100 nm by transmission electron microscopy (TEM) and Dynamic Light Scattering (DLS), respectively. It is found that the structures emerged depended on the relative lengths of the hydrophobic blocks in pluronic. Regarding g2(τ) behavior from DLS measurement, the nanogels showed a high stability in spherical form. Surprisingly, the release profiles showed a sustainable behavior of Cur from this system for drug delivery approaches. In vitro study exhibited that nCur-encapsulated complex micelles increased inhibitory activity against cancer cells growth with IC_50_ is 4.02 ± 0.11 mg/L (10.92 ± 0.3 µM) which is higher than of free curcumin at 9.40 ± 0.17 mg/L (25.54 ± 0.18 µM). The results obtained can provide the new method to generate the hierarchical assembly of copolymers with incorporated loading with the same property.

## 1. Introduction

Hierarchical assembly of the amphiphilic block copolymer, which allows the directly tailor and structure the nanoscale micelles ordered organization has recently drawn great research interest [1,2,3,4]. The emergence of micelle packing with well-design at the higher level complex formulation, which comes from the approaching the concept of biological macromolecules as protein, have proven to be an excellent candidate in drug delivery system field of amphiphilic block copolymer [5,6]. Various supramolecular micelles as the large complex micelles [7,8,9], multicompartment micelles [10,11] and large compound micelles/vesicles [12] have been received via the modification of amphiphilic block copolymer. Among these, we have developed several approaches to obtain complex self-assemblies of amphiphilic block copolymer [4,8], in which specific interactions, mainly hydrophobic–hydrophobic interactions connect the core and shell. Recent reports provide other intricate super micelles that are driven by various noncovalent interactions, such as hydrophobic interactions, electrostatic interactions, and hydrogen bonding, etc. [13,14]. Interestingly, the formation of micelles by complexation of hydrophilic segments with drug counterions, and their cargo structure with the standard interface via hydrophobic–hydrophobic interaction propose the system with such a core packing motif which have attracted significant interest for applications in materials chemistry [15,16].

Self-assemble offers tremendous potential as a method to fabricate the complex micelles in the form of multicore [7,8,9,10,11,12]. In particular, the self-assembled process, the uniform structure with the controllable size of these particles are still the remaining problem [9]. These can be solved by modification the amphiphilic copolymer to improve the thermodynamics of the assembling molecules as the introduction of another component [3,6,11,12] through the control of the natural interaction of each component in solution. However, the co-assembly of this approach is still random and difficult to apply to the general system. Ultrasonication is one of the methodologies can provide the excellent control of environmental conditions of self-assembly. Ultrasonication provides the acoustic wave that can quickly generate nucleation and perfect control the growth and the collapse of bubbles resulting in the synthesis of elementary nanoparticles and support the formation of complex nanoparticles with narrow size distribution and with high yield [17,18]. Although self-assemble by ultrasonic has been emphatically developed in recent decades, this method is not the best strategy in the field of polymeric material [19]. Ultrasonication seldom favors the formation of an ordered assembly because of their low crystallinity. In this work, we show that the dual self-assembled in solution using the interactive nature of compartment together with the help of acoustic waves can achieve not only the remained problem but also can be constructing hierarchical packing micelles.

Pluronic, water-soluble in form of A-B-A or triblock copolymers of poly(ethylene oxide)–poly(propylene oxide)–poly(ethylene oxide) (PEO–PPO–PEO), shows the best self-assembly behavior in water into micelles consisting of a hydrophobic core of PPO and a shell of the solvated PEO [20]. The different in solution properties of poly(ethylene oxide) (PEO) and poly(propylene oxide) (PPO) leads to the versatile self-assembly into micelles, vesicles or the complex one by the changing experiment conditions such as solvents or temperature or additive [20]. It was reported that pluronic F127 micelles altered core size after encapsulating the hydrophobic drug. In addition, via the loading of hydrophobic drug, the gelation temperature of Pluronic F127 was also reduced [21], as similar with another Pluronic member such as P104 [22], P105 [23] and P123 [24]. Besides, the loading hydrophobic drug as paclitaxel (PTX) into Pluronic P123 results in varying morphology from the core–shell structure with the increase of the PTX concentration [9]. The transition from micelles to unilamellar vesicles was also detected after addition of 5-methyl salicylic acid into Pluronic P85 [25] and Pluronic F127 [26]. Therefore, self-assembly becomes a topic of great attractive approach to construct the suitable carriers for biomedical application. However, the micelle system formed by pluronic exhibits the inability to provide sustained drug delivery over more than just a few days [9,20,27]. As a first step to overcome this problem, the modified pluronic was grafted onto several copolymers such as chitosan, heparin, and dendrimer in order to obtain a negative-charged or positive-charged platform resulting in extending its applicability because multi-interactions of bioactive molecules and Pluronic-based platforms were enhanced [27,28,29,30].

Herein, the investigating whether positive charged pluronic-based platform with gelatin generated the complexation with the negatively charged drugs as counterions and hydrophobic–hydrophobic interaction under the support of ultrasonication can form the multicore micelles in aqueous was developed. Moreover, pluronic-conjugated gelatin could be expected to improve biocompatibility with gelatin backbone and stability of the nanocarrier via entanglement effect, effectiveness in as well as well-controlled delivery behavior due to stealth effect by external exposure of PEO domain [31,32,33]. In this study, curcumin was selected as the model because its molecule has both negative charges as well as hydrophobic properties. Beside, curcumin possesses antioxidant, anti-inflammatory, antibacterial, antiviral and anticancer activities [34,35]. The phytochemical could generate a synergic activity with several kinds of anticancer drugs (such as Gemcitabine, Paclitaxel, Camptothecin, and Cisplatin) that was effective against the growth of many cancer cell lines [36,37,38,39,40]. However, Cur is lipophilic nature and poor aqueous solubility as well as low aqueous stability due to its fast metabolism leading to reduce the bioavailability of Cur. Studies on enhancing absorption and solubility via its nanoformulations have been attracting much attention [41,42].

## 2. Materials and Methods

### 2.1. Materials

Curcumin, pluronic F127 (F127, Mw (Da) = 12,600 and CMC = 2.8 × 10^−6^), Pluronic F87 (F87, Mw (Da) = 7700 and CMC = 9.1 × 10^−5^) and p-Nitrophenylchloroformate (NPC) were purchased from Sigma Aldrich (St. Louis, MI, USA). THF tetrahydrofuran (THF), Dichloromethane (DCM), Gelatin (Ge; product no. 1040700500) from porcine skin were purchased from Merck (Darmstadt, Germany). Ethanol and diethyl ether were obtained from Scharlau’s Chemicals (Barcelona, Spain). Mono NPC- activated F127-OH and mono NPC-activated F87)-OH were prepared followed protocol in our previous study [27,40]. Dialysis membranes (molecular weight cut-off/MWCO 14 kDa and MWCO 3.5 kDa cut-off) were supplied from Spectrum Labs (CA, USA).

### 2.2. Synthesis of the Grafted GP Copolymers

In round flasks, 0.25 g of gelatin was dissolved in distilled water at 40 °C under stirring for 1 h and storage below 25 °C for further reaction. 4.5 g of NPC-F127 (F87) -OH respectively dissolved in distilled water at 4 °C and then added dropwise to the gelatine solutions under stirring for 24 h at 25 °C. The reaction mixtures were dialyzed against cold DI water using Cellulose dialysis membrane (12,000–14,000 Mw). After 3 days, the sample was lyophilized to obtain pluronic-conjugated gelatine in white solid form. The grafting efficiency of copolymers was characterized by Fourier transform infrared (FTIR), proton nuclear magnetic resonance (^1^H-NMR), thermal gravimetric analysis (TGA) and critical micelle concentrations (CMC).

### 2.3. Preparation of Cur-Loaded Pluronics and GP Nanogels

In each trial, each amphiphilic copolymer (pluronic F127, pluronic F87 and their conjugated gelatin copolymers) was dissolved into distilled water to reach 10 wt%. 5 mg of Cur was prepared in 4 mL mixture of ethanol and dichloromethane (DCM) at ratio 7:3. The Cur solution was then added dropwise into four copolymer solutions under sonication. In order to prevent the effect of temperature that was generated in this process, a cold-water bath was used. After 5 min sonicating, the obtained solution were evaporated to remove all of the solvent. The drug loading (DL%) and entrapment efficiency (EE%) in polymeric micelles or nanogel were calculated from Equations (1) and (2).
(1)$$\%EE=\;\frac{W_{total\;Cur}-W_{free\;Cur}}{W_{total\;Cur}}\times 100\%$$
(2)$$\%DL=\;\frac{W_{total\;Cur}-W_{free\;Cur}}{W_{total\;Cur}+\;W_{GP}-W_{free\;Cur}}\times 100\%$$

Morphology of the nanogels was observed by TEM (JEM-1400, JEOL, Osaka, Japan). Particle size distribution was determined using dynamic light scattering (DLS).

### 2.4. Release Behavior of Curcumin-Loaded Nanogels

In vitro release profile of curcumin from the formula was investigated following the previous report [34]. Briefly, 1 mL of each cur-nanoformulation was loaded into the dialysis bag (3.5 kD) and then immersed into 5 mL of PBS solution with different pH (pH 7.4 and pH 5.5) at 37 °C. At the predetermined time interval, 1 mL of soaking solution was withdrawn to determine the amount of cur which was released from the visking tube. Then, 1 mL of fresh PBS was put back to vials in order to make sure that there was no change in the final volume of each trial. The release profiles of curcumin were calculated via the standard curve that was constructed by UV–Vis spectroscopy method at 429 nm wavelength.

### 2.5. In Vitro Cytotoxicity

Cytotoxicity test has followed the protocol of Nguyen et al. [43]. Briefly, MCF-7 human breast cancer cell line (HTB-22) at a density of 5000 cells/well was incubated in 96-well plates. After 24 h culture, various concentration of free cur (in DMSO) and its nanoformulation in range 0–10 mg/mL. These wells were further cultured in 48 h before treating with Sulforhodamine B assay for investigation of their preliminary anticancer activity. The percentage of growth inhibition (%I) was calculated according to Equation (3):(3)$$\%I=\left(1-\;\frac{{\mathrm{OD}}_{t}}{{\mathrm{OD}}_{c}}\right)\times 100$$
in which ODt and ODc are the optical density value of the test sample and the control sample, respectively. Camptothecin (Calbiochem) was used as a positive control, whereas DMSO was use as a negative control.

## 3. Results and Discussion

### 3.1. Characterization of the Amphiphilic GP Copolymers

Gelatin is gained much attention in tissue engineering because of its high biocompatibility and biodegradability. Due to the presentation of Arg-Gly-Asp peptide sequences, gelatin can promote cell adhesion [44]. Also, gelatin is a positive-charged at physiological condition [45]. These reasons could lead to a highly attractive interaction to normal or cancer cells.

The synthetic route to the amphiphilic gelatin copolymer (GP) was performed by a grafting to method, in which hydroxyl groups of Pluronic was first modified with p-NPC to make possible coupling reaction between Pluronic and the amine groups (-NH_2_) on gelatin backbone, as similar with our previous study [46]. Due to the commonality in chemical structure of F87 and F127, the scheme in Figure 1 illustrating a two-stage synthesis of GP from F127 was also used to demonstrate for F87. Through the examination of FTIR spectrum (data is not shown), the absorption of amide I (-C=O) and of amide II (>N-H) in GP-F127 show the bathochromic shifts from 1647 cm^−1^ and 1548 cm^−1^ to 1651 cm^−1^ and 1653 cm^−1^, respectively, as compared with pure gelatin spectrum. ^1^H-NMR (500 Hz, D_2_O) was conducted to verify whether the pluronic was successfully grafted to the gelatin backbone. As shown in Figure 2 and compared with another study [46,47,48], ^1^H-NMR spectrum of GP-F127 includes both proton signals of the amino acid-forming gelatin and proton signals of F127. The peaks at 0.87 ppm is assigned to the methyl protons resonance of Leucine (Leu), Valine (Val) and Isoleucine (Ile). The resonance signal at 1.17 and 1.34 are derived from proton of methyl groups in Threonine (Thr) and Alanine (Ala). The signal that appears at 1.61 ppm is attributed to proton in Arginine (Arg). Furthermore, several proton signals at 2.66 ppm and 2.93 ppm are ascribed to methylene groups of aspartic acid (Asp) and lysine (Lys), respectively. The resonance proton signals of aromatic ring of phenylalanine are also presented in the spectrum as the 7.20 ppm. Along with the presence of protons signal of amino acid forming the gelatin backbone, ^1^H of protons signal of PPO blocks of F127 including methyl (-C**H**_3_), methylene (-C**H**_2_-) and methine (>C**H**-) are apparent at 1.12 ppm, 3.40 ppm and 3.55 ppm, respectively. In addition, the strong resonance signal at 3.64 ppm is regarded as proton of -C**H**_2_CH_2_- composing in PEO of F127. It is also the same pattern with GP-F87. These obtained results reveal the strong evidence for a successful grafting method by which pluronic conjugated to amino groups of gelatin via urethane linkages.

Moreover, the thermogravimetric analysis (TGA) test further confirmed this conclusion. Figure 3 shows the TGA curve of amphiphilic gelatin copolymer (solid line), pure gelatin (dashed-dotted line), and pure pluronic (dashed line). For pure gelatin, the first weight loss at approximately 120–140 °C is due to the removal of water; the second weight reduction started at about 200 °C, revealing the denature of gelatin structure through the broken intermolecular interaction between amino acids forming the 3D structure of gelatin [49]. There are nonsignificant points in thermal behavior of both kinds of pluronic, F127 (Figure 3a) and F87 (Figure 3b). Both show the single step of weight loss between 300–410 °C. This is consistent with previous reports related the decomposition of Pluronic polymer [50]. TGA curves of all amphiphilic gelatin copolymers (GP-F87 and GP-F127) are exposed the greatest reduction in mass in the temperature range of 250–420 °C, which are induced by the disintegrating intermolecular interaction as well as the partial interrupting molecular structure of GP. This implies the presence of gelatin in these graft copolymers, which can explain the difference in thermal decomposition behavior. In other words, the introduction of gelatin helps to increase the thermal stability of pluronic. Thermal decomposition temperature of both pluronics are around 410 °C while GP and pure gelatin are around 600–800 °C. Regrading to these thermal behavior of all samples, grafting efficiency was calculated via the percent weight loss of GP samples at 420 °C at which pluronic backbone was completely decomposed. Based on the ratio between weight loss of pure gelatin and its remained mass in GP at 420 °C, the weight loss of grafted pluronic on gelatin backbone was calculated, following the report of Kang et al. [51]. The graft yield of GP-F127 was 49.45% whereas yielding of 54.7% was achieved in case of GP-F87 as seen in Table 1.

The amphiphilic GP copolymers consisting of the hydrophilic parts (gelatin chain and PEO segments) and the hydrophobic PPO segments provide an opportunity to examine their self-assembly behavior in aqueous solutions. The micellization of amphiphilic GP copolymer in aqueous solution was investigated using pyrene as a fluorescence probe. It was reported that when micelles form in aqueous solution, the hydrophobic pyrene can be encapsulated into hydrophobic PPO microdomains that causes an increment in the fluorescence intensity ratio of bands (I1 nm/I3 nm as 384 to 373 nm) at the pyrene fluorescence emission spectrum. Figure 4a shows the critical micellization concentration (CMC) of GP-F127 is 117 ppm (0.117 mg/mL) while GP-F87 is 186 ppm (0.186 mg/mL) as seen in Figure 4b. According to the higher ratio PPO/PEO in Pluronic F127 (65/200) than in Pluronic F87 (39/122), the abatement in CMC values of GP-F127 is prevised result. In addition, the CMC value of amphiphilic gelatin copolymer is higher than pure pluronic (CMC value of Pluronic F127 (0.035 mg/mL) and Pluronic F87 (0.071 mg/mL) [52]). The results also indicated that the introduction of gelatin leads to the superiority of hydrophilic segments for which the CMC values of the GP is greater than that of pure ones. The difference in CMC values of GP and pluronic copolymers could lead to a peculiar behavior in hydrophobic drug loading efficiency.

### 3.2. Optimization of Curcumin Loading Content

In the manner of drug delivery system, entrapment efficiencies (EE) and drug-loading efficacy (DL) are a crucial parameter that directly affects the therapeutic efficacy of the system [53]. In this study, the curcumin loading ranged from 5 to 20 wt% while variations such as the final concentration of carriers, the condition of ultrasonication (time, power as well as amplitude) were kept as constant. As shown in Figure 5a, the EE of all carriers increase to highest values when curcumin content used at 10% with respect to copolymers content. And then the significant reduction was observed after further feeding curcumin (15–20 wt% of concentration) at all studied cases. In order words, the DL of the all carriers significantly increased at formulations loading from 5 to 10% of curcumin and slightly increased after the point as seen in Figure 5b. Therefore, drug-releasing behaviors were further studied using the 10 wt% of curcumin. The EE of pluronic micelles are much lower than that of its GP nanogels around 10–15 wt%. These results could be affected by both interactions of curcumin with hydrophobic PPO segment and characteristic of gelatin structure.

In fact, Table 2 indicated that two copolymers formed negative-charged micelles but pluronic F127 possesses a higher hydrophobicity or lower hydrophilic and lipophilic balance (HLB) resulting in an increment in its curcumin loading efficiency as compared to pluronic F87. While GP-F87 with a high CMC value as well as a negative-charged structure (−7.9 ± 0.1 mV from Zeta potential) and the higher hydrophobic pluronic F127 performed a lower DL and EE as compared to GP-F127, a positive-charged copolymer (+7.67 ± 0.21 mV from Zeta potential). This confirms that both electrostatic and hydrophobic interactions impacted DL and EE of GP nanocarriers. These results are fascinating to develop several Pluronic-based platforms for improving drug delivery efficiency which has matched a trend in the modification of pluronics for drugs delivery systems [54].

### 3.3. Characterization of Cur-Loaded Pluronics and GP Nanogels on Morphology and Stability

For encapsulating nCur, it is vital to select a mixture of an organic solvent for dispersion of curcumin and partial compatibility with amphiphilic copolymers. In some investigations, ethanol showed a fair estimate of the proportions of its keto-enol forms of curcumin [55]. Dichloromethane (DCM) is one of the best solvents to use in extraction as well as in nCur processing [56,57]. Regarding our screening experiments, the co-solvents (ethanol and DCM) were used to produce nCur in amphiphilic copolymer solutions under ultrasonic-assisted condition. Transmission electron microscopy (TEM) was employed to study the morphology, whilst size and dispersion of nanoparticles was examined via dynamic light scattering (DLS). Figure 6a demonstrates that nCur/F87 have spherical morphology with hydrodynamic diameter of about 65.1 nm (Figure 6(a1)). Upon the self-assembly of pluronic copolymer, curcumin molecules was encapsulated in the hydrophobic regions of formed nanogels in cases of study. The hydrophobic interaction between the hydrophobic domain of Pluronic and the aromatic phenols of curcumin molecules to form intermolecular complexes may be the main inducing force for the formation of nanogel. From TEM images, Figure 6a,c show nCur-loaded micelles or nanogels formed ranging from 10 to 100 nm in which a distinguishable presence of nCur is apparent in these images. However, Figure 6b,d show the slightly different morphology of nCur-encapsulated pluronic F127 micelles or GP-F127 nanogels. In Figure 6b, the core–shell nanoparticles appear in a wide range from 20 to 100 nm in diameter. The behavior could derive from the encapsulation of hydrophobic (nCur) in hydrophobic domains of pluronic molecules. In the case of GP-F127 encapsulation, the diameter of these spheres is ranging from 40 to 70 nm. Moreover, a higher density of nCur aggregated in GP-F127 nanogels as seen in Figure 6d. This phenomenon could be produced by the characteristic of nCur and GP-F127 copolymers such as hydrophobicity and electrostatic interaction with each other. For further studies, size distribution of the the nCur-loaded GP-F127 were obtained from DLS indicating their homogeneous dispersion in colloidal solution as seen in Figure 6(d1) when compared nCur-loaded micelles or GP-F87 nanogels (Figure 6(a1–c1)). Moreover, in Figure 6(a2–d2), the normalized intensity autocorrelation functions g2(τ) fit to a stretched exponential forms which could indicated that the nanoparticles produced at the spherical micelles without free unimers [58]. The results could confirm that the nanoparticles showed a high stability in colloidal solution without any dissolution or aggregation.

Regarding to the obtained results, the self-assembling process of GP-F127-nanogels and nCur could be demonstrated in Figure 7a leading to improve curcumin-loading efficiency in the nanogel. Moreover, during self-assembly process, the hydrophobic segments on pluronic as well as gelatin structure in turn to aggregate to reduce the interaction with the aqueous environment and increase interaction with the inner loaded ncur. The phenomenon could be well-observed regarding to characteristic in fluorescent adsorption of curcumin at 425 nm excited wavelengths. The colloidal GP-F127-nCur nanogels were prepared in microscope glass slides without any label agents and capped with coverslips. Fluorescent images of the Andor Dragonfly confocal microscope indicated morphology of the nanocurcumin-encapsulated cationic and amphiphilic GP-F127 copolymer (as seen in Figure 7b,c).

### 3.4. In Vitro Drug Release

Curcumin release behavior from pluronic F127 micelles and GP-F127 nanogels were performed under in vitro conditions at acidic and physiological pH mediums. It is established that cancer cells are acidic. Thus, release studies were performed at pH 5.5 and pH 7.4. The Cur release studies were carried out for 120 h. Cur-loaded pluronic micelles and nanogels show a sustained released behavior at pH 5.5 which reach to 75% after 120 h as seen in Figure 8. Moreover, the rate is much faster than that of pH 7.4, which is only approximately 20% at the same time. The contrast in release curves in both pH values could be explained by the effect of pH on ionization of a hydrophobic molecule like a case of Ibuprofen-loaded pluronic-F127 [58]. A higher pH leads to stronger ionization of the molecule that loosens hydrophobic core of pluronic resulting in the increased release rate of the loaded molecules. A slower release rate could see in GP-F127 at both pH points which is assigned with electrostatic of curcumin and positive-charged gelatin chains. The conjugation of the gelatin results in extending the applicability of Pluronic F127 in drug-controlled delivery.

### 3.5. Cytotoxicity of nCur-Loaded GP Nanogels

To assess the potential of anticancer cell growth of nCur-loaded GP nanogels for further applications, the in vitro cytotoxicity of the free Cur in DMSO and the nCur-loaded nanogels evaluated using human breast cancer cell lines (MCF7). The in vitro half-maximum inhibitory concentration (IC_50_) is the quantitative parameter to evaluate the cell toxicity induced by samples that were calculated from Figure 9. In the figure, the sample of free curcumin performs low toxicity against MCF7 cancer cell growth with IC_50_ value at 9.40 ± 0.17 mg/L (25.54 ± 0.18 µM). The value is also approximately equal to Paola’s report, which is 29.3 ± 1.7 μM in MCF-7 [59]. It is interesting that the IC_50_ of the nCur-loaded nanogels sample was calculated at a significantly low value that was 4.02 ± 0.11 mg/L (10.92 ± 0.3 µM). The result indicated a higher activity against the cancer cell growth as compared with free curcumin. The performance could be explained various reasons such as low stability of curcumin in physiological condition, efficient protection and well-controlled delivery of the bioactive molecule from GP nanogels. Moreover, it could derive from the characteristic of the gelatin-based nanogels such as its Arg-Gly-Asp sequences promoting cell adhesion and highly electrostatic interaction of the positive-charged nanogel with cancer cell membrane [44].

## 4. Conclusions

By exploiting the properties of loading agent and a grafted copolymer, core–shell GP-F127-nCur nanoparticles were fabricated by dual hydrophobic and electrostatic interactions of the positive-charged amphiphilic GP-F127 copolymer and a negative-charge and hydrophobic bioactive curcumin in organic co-solvents under ultrasonic assistance. The method produced spherical core–shell nanogels of nCur-aggregated GP-F127 ranging from 40 to 70 nm in diameter. The dual interaction of the GP-F127 copolymer with nCur contributed to increasing the curcumin loading efficiency by 10% as compared to pluronic F127 only. It is interesting that the nCur-encapsulated nanogels conducted higher activity against the cancer cell growth as compared with free curcumin due to a well-controlled delivery of curcumin and interaction of the nCur-loaded nanogel with the cell membrane. Due to this excellent performance, this fabricating method could be extensively studied with anticancer drugs to produce several synergistically dual drugs delivery systems for cancer therapy.

## Figures and Tables

**Figure 1 polymers-11-00814-f001:**
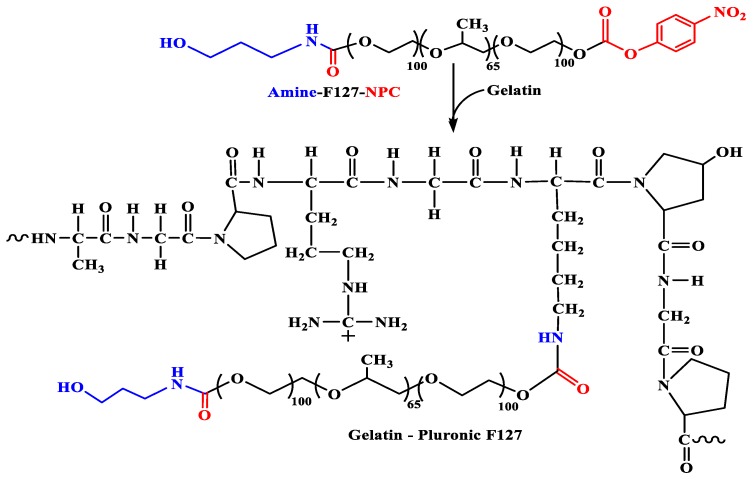
Synthetic scheme of GP copolymer.

**Figure 2 polymers-11-00814-f002:**
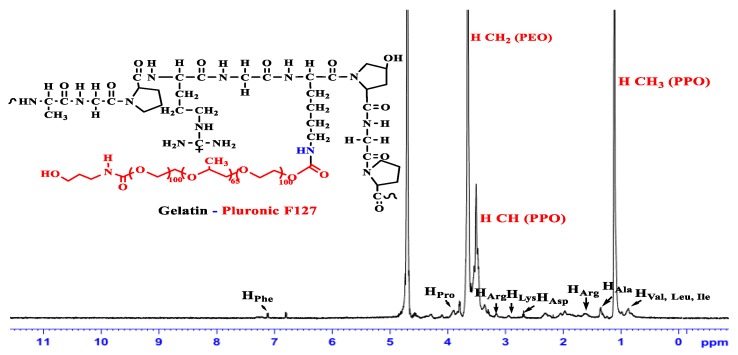
^1^H NMR spectrum of pluronic F127-grafted gelatin copolymer in D_2_O.

**Figure 3 polymers-11-00814-f003:**
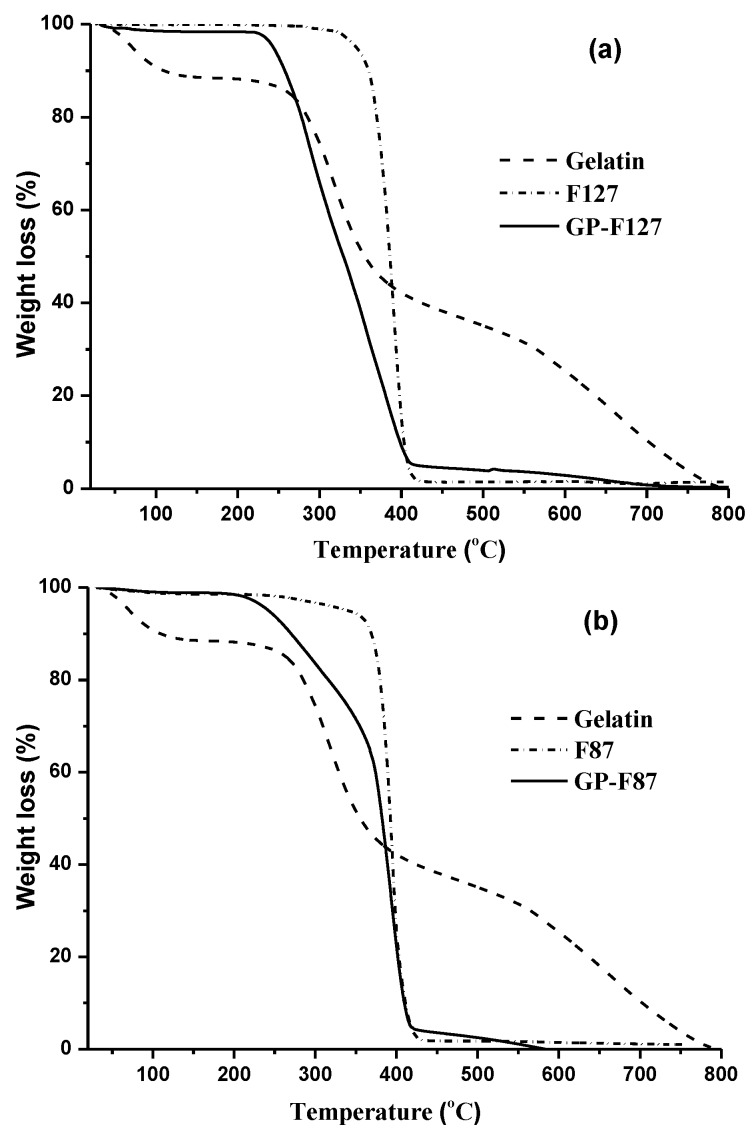
TGA thermogram of gelatin, pluronic F127 (**a**), pluronic F87 (**b**) and its grafted gelatin.

**Figure 4 polymers-11-00814-f004:**
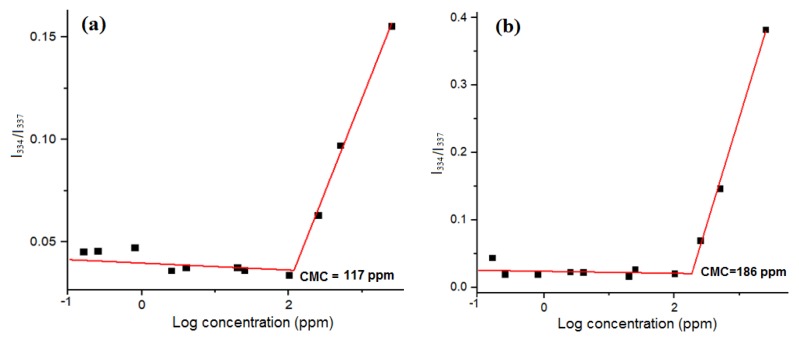
Fluorescence intensity ratio of bands (I1/I3 of pyrene as a function of the logarithm of the copolymer concentration that indicated CMC values with the intersection of the tangents: (**a**) GP-F127 and (**b**) GP-F87.

**Figure 5 polymers-11-00814-f005:**
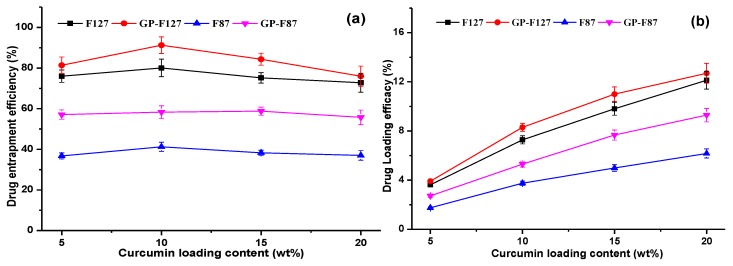
Entrapment efficiencies (**a**) and drug-loading efficacy (**b**) of pluronics and its grafted forms (GP-F127 and GP-F87).

**Figure 6 polymers-11-00814-f006:**
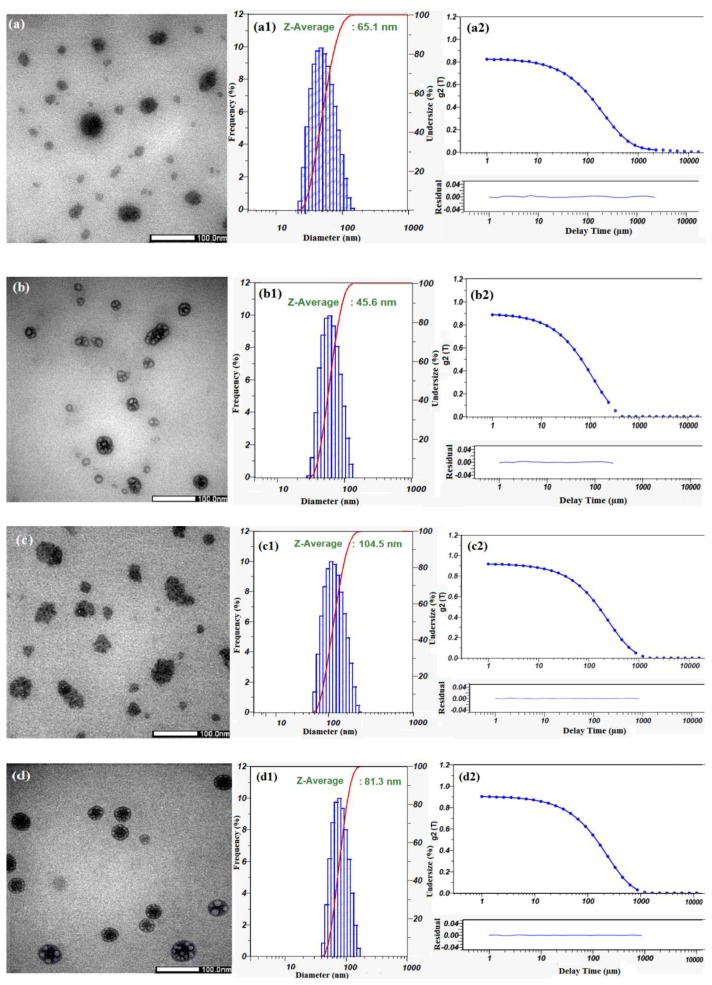
TEM images of pluronic and GP copolymers encapsulated (or dispersed) nCur: (**a**) pluronic F87, (**b**) pluronic F127, (**c**) GP-F87 and (**d**) GP-F127.

**Figure 7 polymers-11-00814-f007:**
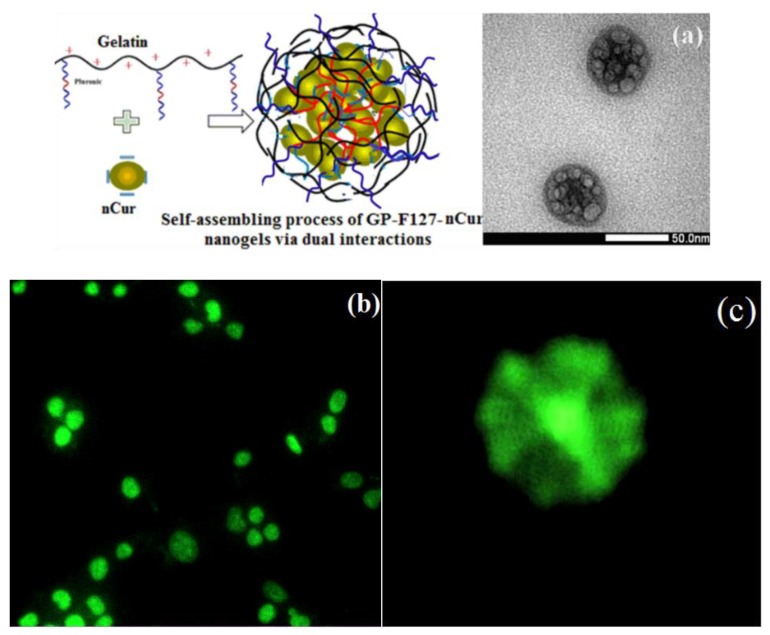
Demonstrating self assembling process of GP-F127 and nCur via dual interactions (**a**), Andor Dragonfly confocal microscope of GP-F127-nCur under 100× objective (**b**,**c**).

**Figure 8 polymers-11-00814-f008:**
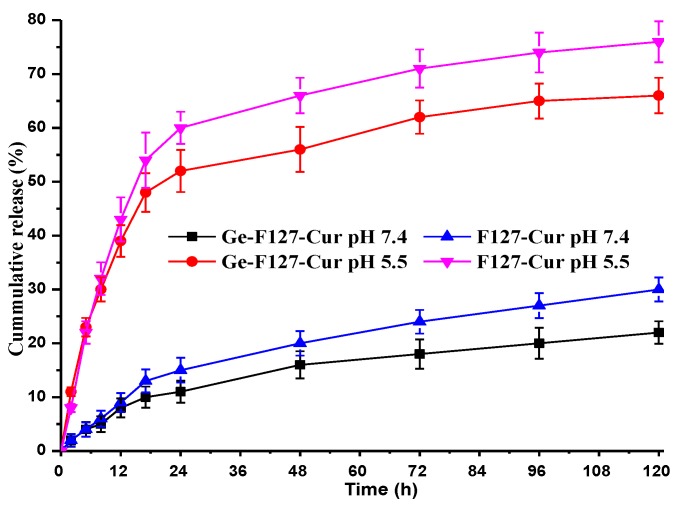
Release profiles of the encapsulated curcumin in Ge-F127 and F127 nanoparticles at two physiological conditions, pH 7.4 and 5.5.

**Figure 9 polymers-11-00814-f009:**
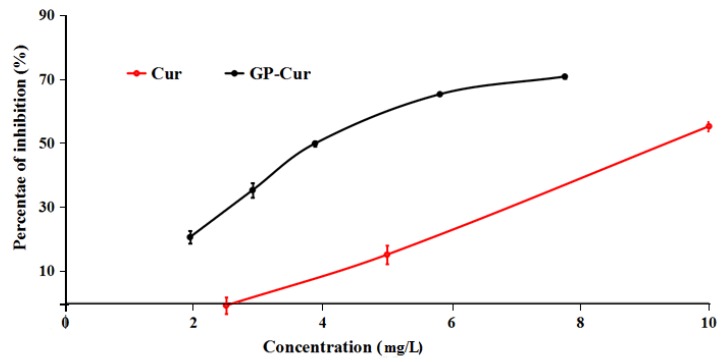
Cytotoxicity behavior of free Cur- and nCur-encapsulated GP-F127 samples on the inhibition of cancer cells growth.

**Table 1 polymers-11-00814-t001:** Polymer composition of the grafted copolymers from TGA curves.

Sample	Gelatin (%)	Pluronic (%)	Grafting Yield (%)
GP-F127	10.10	89.90	49.45
GP-F87	9.22	90.78	54.70

**Table 2 polymers-11-00814-t002:** Zeta potential of pluronic and pluronic-grafted gelatin copolymers.

Pluronic	HLB	Zeta (mV)	GP	Zeta (mV)	GP-Cur	Zeta (mV)
F127	22	−22.67 ± 0.21	GP-F127	7.67 ± 0.21	GP-F127-nCur	−24.20 ± 0.53
F87	24	−29.77 ± 1.11	GP-F87	−7.9 ± 0.1	GP-F87-nCur	−31.43 ± 0.74

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
