# Peer review of "Dual Interactions of Amphiphilic Gelatin Copolymer and Nanocurcumin Improving the Delivery Efficiency of the Nanogels"

_polymers, 2019, doi:10.3390/polym11050814_

Round 1
Reviewer 1 Report
After revision the manuscript is now suitable for publication.
Reviewer 2 Report
I think the current version seems OK for me. Need to re-check the ref. 32, the author name is wrong.
This manuscript is a resubmission of an earlier submission. The following is a list of the peer review reports and author responses from that submission.
Round 1
Reviewer 1 Report
The manuscript describes the self-assembly nanogels that improve the efficiency of curcumin via amphiphilicity and electrostatic interaction. While the authors have carefully made a few characterizations, these tests are routine and lack some in-depth analysis and explanation of the results. The whole work is relatively complete but lacks innovation include the method of self-assembly through amphiphilic and electrostatic interactions. I think this paper can only be re-considered after fully addressing the major concerns as following:
1. The authors could add the following references which would again increase the interest to general hydrogel material readers: Journal of Controlled Release 2018,273, 160-179; Journal of Materials Chemistry B 2019, 7, 709–729; Advanced Drug Delivery Reviews, 2010, 62, 83-99.
2. In the introduction part, the authors need to clarify the significance of this work compared with many other nanogel systems. Why use Pluronic to make the nanogel is interesting?
3. In Fig 5, are the formed micelles thermodynamic stable? Need to discuss this in the paper.
4. As the authors are non-native English speakers, there are quite a few small errors and non-standard usages of words and phrases. A thorough revision of the language is required before the manuscript can be accepted.
Reviewer 2 Report
The manuscript describes just another particle formation for drug delivery. Here, particles of gelatin and two kinds of Pluronics were prepared. The obtained polymers are not well characterized, e.g. there is no quantification of the composition etc. It is not clear what the sense of TGA measurements is for drug delivery prurposes. The prepared particles have a very broad size distribution. TEM is not the best method to characterize the size. The authors should do light scattering studies. Further the authors should check the abbreviations throughout the manuscript. At some points they just appear. The bad English makes the manuscript difficult to read.